# A New Method for Gaining the Control of Standalone Underwater Sensor Nodes Based on Power Supply Sensing

**DOI:** 10.3390/s21144660

**Published:** 2021-07-07

**Authors:** Daniel Rodríguez García, Juan A. Montiel-Nelson, Tomás Bautista, Javier Sosa

**Affiliations:** Institute for Applied Microelectronics (IUMA), University of Las Palmas de Gran Canaria, 35017 Las Palmas de Gran Canaria, Spain; drgarcia@iuma.ulpgc.es (D.R.G.); montiel@iuma.ulpgc.es (J.A.M.-N.); jsosa@iuma.ulpgc.es (J.S.)

**Keywords:** battery sensing, sensor network failure, standalone sensor node, underwater sensor node, precision aquaculture, offshore fish farm

## Abstract

In this paper, a new method for gaining the control of standalone underwater sensor nodes based on sensing the power supply evolution is presented. Underwater sensor networks are designed to support multiple extreme scenarios such as network disconnections. In those cases, the sensor nodes involved should go into standalone, and its wired and wireless communications should be disabled. This paper presents how to exit from the standalone status and enter into debugging mode following a practical ultra-low power design methodology. In addition, the discharge and regeneration effects are analyzed and modeled to minimize the error using the sensor node self measurements. Once the method is presented, its implementation details are discussed including other solutions like wake up wireless modules or a pin interruption solution. Its advantages and disadvantages are discussed. The method proposed is evaluated with several simulations and laboratory experiments using a real aquaculture sensor node. Finally, all the results obtained demonstrate the usefulness of our new method to gain the control of a standalone sensor node. The proposal is better than other approaches when the hibernation time is longer than 167.45 μs. Finally, our approach requires two orders of magnitude less energy than the best practical solution.

## 1. Introduction

In general, the practical deployment and usage of an underwater sensor network in real applications makes mandatory at least the isolation of their sensor nodes. The objective is to avoid the potential hazards for the electronic devices, that is, to keep the electronics away from contact with humans, wildlife, gaseous, or liquid mediums, or other potential problems like pressure, temperature, or humidity [1].

The isolation condition of the sensor nodes promotes the research of wireless solutions like ultrasonic or with RFID, for example, or specific connectors for wired communications [2]. In general, when a failure arises in communications the sensor nodes go into standalone mode. Despite of this failure, each sensor node still remains executing its programmed measurement schedule. Therefore, it continues acquiring data that become locally stored when there is memory available. This scenario defines a very common corner case in sensor networks.

On the other hand, the restoration of the failure in the sensor network is not an easy task. Obviously, it is desirable in a short period as the failure occurs until it is fixed. However, in practical applications, restoring the communications depends on the environmental conditions and the distance to the location of the sensor node. For example, in underwater sensor networks, the diving operations to repair the network depend on ocean waves, tides, and wind conditions. In the cases of forest fires and seismic or volcanic detection applications, the distance between sensor nodes and the control center requires programming the movement of an operator over a broad area.

Once the sensor node is replaced and the communication reestablished, the faulty sensor node is back to the laboratory or the repair center. Then, the first step to perform is to download the acquired data after the failure. However, the isolation of the sensor node with broken communications introduces a prior task, which is to gain the access and the control of the sensor node.

In addition, in underwater applications, their printed circuit boards and devices are also protected using a coverage of non-conductive resins or similar materials. Certainly, most of the sensor nodes design includes debugging access points following standards like JTAG [3]. However, the required security systems to keep the intellectual properties safe avoid the access to the sensor node using traditional debugging tools. The traditional way to gain access and to take control of the isolated sensor node is basically to wait for the end of the programmed measurement schedule. Once the schedule finishes, the device activates its internal debugging terminals. Moreover, it is remarkable that the sensor nodes powered by batteries do not allow to use a solution like one-wire [4] or single-wire [5] communications for finishing the execution of its schedule.

This paper proposes to use the charging process of the battery included in the standalone sensor node for gaining control over it. In relation with this approach, power consumption management in sensor nodes and networks is a hot topic that has been also widely studied in various directions. Multiple models have been developed to define the behavior of batteries in a wide variety of applications. For example, a circuit-based model [6] and a formulation model [7] have been proposed for Lithium (Li) batteries. An extensive overview and presentation of battery modeling techniques, with a convenient taxonomy of them, has been also published [8].

In addition, and directly related to the quest for reduction of consumption, algorithms have been published in the literature for adapting the workload demand for better energy management [9]. Nevertheless, other works have made several relevant contributions for sensor networks by considering the effect of duty cycling and buffering to harness the so-called regeneration effect of batteries as an energy-saving mechanism [10,11].

The regeneration effect affects devices with pulsed or intermittent consumption [12]. This behavior corresponds to the operation of the device dealt with in this work. The details and the study of the energy behavior in these systems have been presented with a detailed electrochemical battery model [13] using a stochastic model of the device. In other studies [14], attention has been paid to a simple battery on–off switching technique called pulsed discharge control as a means to extend the lifetime of the battery.

On the other hand, in the research concerning underwater sensor networks, the contributions are mainly focused on reducing the required communications power consumption and optimizing the routing protocols [15]. The applications in the literature are mainly related to monitoring environmental parameters, aquaculture industry activities, and aquatic wildlife [16].

From the deployment point of view, two main scenarios exist: In the first one, the sensor network is partially located in the aquatic environment. For instance, this is the case of a system for water quality monitoring and measurement in rivers or aquaculture tanks [17,18]. Another example is a feeding monitoring sensor network for aquaculture tanks, which is presented in [19]. In both cases, their approaches involve underwater wired sensors to monitor the tank with a hub on the water surface which concentrates all the data collected and the communications with an Internet server.

In other scenarios, where data have to be collected from a diversity of elements, the complete sensor application is underwater [20]. Depending on several design variables such us the mobility of the sensed target, the energy consumption requirements or the nature of the measurement, the sensor solution could include a network, or a set of sensor nodes without communication capabilities among them.

A critical case, for example, is the EEG acquisition for dolphins [21]. Due to their high mobility and wide range of movements, the sensor node is not optimal in terms of power consumption due to its Bluetooth communications needs. In the worst case, if the underwater sensor network fails [22] or when the sensor node application requires a solo deployment [23], the sensor nodes are powered by batteries and completely isolated.

On the other hand, the research presented in [24] studies a detection and isolation method when an underwater observation network fails. The authors propose a hierarchical topology based in hubs and sensor nodes to detect high-impedance and open circuits faults in wired underwater sensor networks. In a similar way, the authors of [22] define a mixed wired–wireless deployment strategy for offshore fish farm cages taking into consideration the faults of the network.

At the sensor node level, the work in [25], for instance, studies in detail the energy consumption of a wireless sensor node and proposes stochastic modeling for a given experiment schedule. As a result, the lifetime of the sensor node is estimated for a battery-based sensor node with wireless communication capabilities. In a similar way, a Markov process is used to model the energy consumption of a wireless sensor node in [26]. The design and power management of a secured wireless sensor node for environmental monitoring is detailed in [27]. This study focuses the attention of the reader on the power consumption analysis of each sensor node subsystem when it is built with commercial off-the-shelf (COTS) peripherals. Those recent studies are focused into the regular operation of the sensor node.

At all the design levels in the literature, when a failure appears the research is focused in minimizing the impact of the problem. All the researchers assume that a faulty sensor node must be replaced. Once the network problem arises, the sensor node goes into autonomous mode. In this case, the wireless sensor node works as a standalone sensor node. This means that the node remains working with the programmed measurement schedule but without communication. If the communication cannot be reestablished, the sensor node saves the so-called off-line measurements.

In order to access and download those valuable measurements, the sensor node must be manipulated in a specialized laboratory. In a similar way, this problem also arises in standalone sensor nodes when a programmed experiment must be interrupted due to an ending user decision. The wireless module failure, the sensor node isolation from the water, and its disabled terminals conjointly define a non-trivial scenario for taking the control of the sensor node in the laboratory avoiding non-destructive techniques.

In order to gain the control of the sensor node in such a scenario, an approach has been established. The main contributions of this approach, presented on this paper, are the following:This paper describes how to implement a practical method for gaining the control of a standalone sensor node for ultra-low power design.The sensor node self battery sensing is studied and discussed with practical considerations.The proposed approach implementation is evaluated and validated under several real underwater applications.

The remaining of this paper continues with Section 2, which describes the operation and execution modes of the sensor node, the reasons for its isolation, the need to be able to take out the device from it and how this is achieved. The implementation details from a practical point of view of the proposed software approach is presented in Section 3. In the next one the results obtained in a wide set of experiments are shown, evaluating the errors along the methodology and also testing the solution, including comparisons with similar approaches found in the literature and commercially available. Finally, in Section 7, the most relevant conclusions are enumerated.

## 2. System Description

The objective of a sensor node is to measure a set of target variables at specified time instants or ranges. In general, a complete set of timed measurements is called an experiment. Figure 1 presents a simplified execution scheme of a sensor node to explain a bit more about this term in this context. Basically, the experiment is an iterative process that executes two tasks sequentially. One of the steps configures the on-board instruments and executes the measurement. The other one keeps the whole system sleeping until the next iteration.

After the experiment is finished, the system waits for an operator to download the data. Furthermore, if the sensor node has communication capabilities, there is at least one transmission and reception method that interacts with the steps described above. This execution scheme belongs to the so-called autonomous or standalone operation mode.

### 2.1. Practical Implementation

From an economical point of view, the isolation obtained with the sensor node container or capsule is of great concern. In order to avoid the usage of expensive connectors, wireless solutions provide an economical approach in comparison with wired methods. Unfortunately, despite the high isolation degree obtained, intermediate or prolonged exposure to the marine environment may cause the presence of humidity inside the capsule. Therefore, a second protection line against this problem is required. The solution is to provide a fence to the circuits of the sensor node using water resistant resins or silicones.

In addition, the sensor node turns off any voltage across its terminals, even within the capsule; otherwise, if these voltages are not canceled, an electrolysis process occurs between terminals when humidity appears. Its effects are devastating in a double dimension. On one hand, the partial or total loss of metal in the affected terminals. Furthermore, the discharge of the battery would also occur in such a situation. Eliminating the voltage on these terminals is achieved by effectively disabling them and, with this, their functionalities.

On the other hand, power consumption is another important key issue in the research of sensor node networks. Every design decision implies a consequence in terms of power consumption. For example, selecting a microprocessor working mode or another in the Sleep or Exe steps shown in Figure 1 substantially affects the power consumption. Obviously, the reduction in power consumption is greater as the system turns off more and more functionalities.

In summary, an underwater sensor node is an ultra-low-power smart system highly isolated from its environment, in which these specific features reduce its communication and design for testability capabilities.

### 2.2. Other Operation Modes

In addition to the tasks previously mentioned, a sensor node must also perform other tasks, such as to reschedule or reprogram its experiment, to calibrate an instrument, and to download or erase the data or system logs from memory, among others. These additional tasks require the use of other internal operational behaviors, for which a mode with complete access to all the resources in the system should be available. The implementation of several security levels on the sensor node makes necessary different supervisor modes. However, all of them follow a master–slave architecture, where the sensor node acts as slave and a human operator or supervisor, or a monitoring system works as master.

The isolation and ultra-low power design implementation introduces some critical corner cases in the switching mechanism between operating modes. Certainly, going from supervisor to automatic mode is quite simple. For this, the master only needs to command the slave to go into the other mode; at that time, the experiment is started and executed. However, the reverse path is not so direct. It is obvious that the waiting state in the autonomous mode is one of the entry points to the supervisor mode. Nevertheless, there are also other scenarios which require to enter into supervisor mode. This is the case, for instance, when error conditions arise or a user orders to abort an experiment.

In this paper, the influence of the triggering procedure is studied in detail, switching from autonomous to supervisor modes of a sensor node in terms of energy consumption and its implementability to a real underwater sensor node.

### 2.3. Discussion about Triggering Mechanisms

It should be observed that this triggering mechanism is used by a maintenance operator in a laboratory. The first objective of this mechanism is to download the off-line acquired data. The second objective is to reduce to the minimum the manipulation of the standalone underwater sensor node to download its data and logs. In order to guarantee the trigger mechanism if a wireless solution is adopted, the selected wake up unit must be exclusively dedicated to this task. Because the usage of the wake up module is restricted to laboratory operations, all wireless wake up technologies are suitable for this purpose. The usage of a specific wireless wake up approach is only limited by its power consumption requirements and by the antenna dimensions, which must comply with the underwater sensor node design constraints.

Identified the design considerations of an underwater sensor node, in this section, the triggering procedure for switching to supervisor mode is discussed. There are three possible solutions for implementing the triggering mechanism: wireless wake up module (WM), pin interruption (PI), and our proposal, which is based on a software approach (SA). Table 1 summarizes the benefits and weaknesses of these three methods.

Having as a primordial objective to guarantee the isolation of the underwater sensor node, the wireless solution is the most robust. Obviously, this approach requires the use of a radio frequency wake up circuitry and an antenna. In addition to the area increase on the printed circuit board (PCB) and the antenna orientation problems [28], its power consumption must be taken in consideration [29], and, in particular, its quiescent current [30].

When a pin interruption (PI) is used [31], the PCB extra area and antenna orientation problems do not exist in comparison with the WM approach. Moreover, the wake up latency is also lower than using a WM, but the physical manipulation of the device is more complex when the PI solution is used. Furthermore, the activation of the auxiliary circuitry to support the wake up interruption from an external pin increases the power consumption of the sensor node, especially in deep sleeping modes [32].

Finally, the software approach (SA) proposed consists in evaluating an internal state or variable of the sensor node to trigger the switching into supervisor mode. For example, the SA stops the experiment when a fatal error arises, as it might happen because of a failure in an on-board instrument or the battery. Obviously, this solution does not require extra PCB area. However, by its nature it is mandatory to execute a piece of code for making checks with the CPU in normal running mode. This implies that the triggering may not performed with the CPU in sleeping mode. In that case, the total power consumption depends only on the normal CPU running mode throughout the cycles involved in those checks.

Based on the discussion above, it is concluded that in terms of ultra-low power and isolation features there does not exist a specific winning solution. Furthermore, the WM is considered an isolation upgrade over PI as the underwater sensor node must activate a pin interruption for attending the wireless wake up module. Anyway, if a failure arises in the wireless circuity or antenna, it is necessary to open the sensor node container.

## 3. Proposal of a Practical Software Approach

One of the milestones in this research is to develop a trigger mechanism to switch the underwater sensor node from its autonomous mode to its supervisor mode without a significant impact on the performance, especially in terms of energy. A software approach requires at least a threshold function to determine or not the triggering. In this paper it is proposed to use the evolution of the battery charge to trigger the mode switching.

Nowadays, sensor nodes based on batteries execute a software assertion mechanism for checking battery failures. This sanity evaluation consists in examining the power supply within the immediate instructions executed after each CPU wake up. This implies the usage of a low voltage threshold detector and the measurement of the power supply voltage.

Figure 2 shows the internal programming and charging connector used in the underwater sensor node [22]. In this research, this sensor node is used to implement and test the approach followed. As can be seen in the figure, the connector is similar in layout to that of a microSD, and in the figure, the power supply terminals VDD and GND can be identified. The Serial Wire Debugging (SWD) terminals are labeled as CLK and DIO. In addition, this connector gives access to the microcontroller Reset terminal and the General Purpose Input Output (GPIO) identified as PTE30.

Disabling the reset terminal (RST) is mandatory in this sensor node for underwater applications. By default, the RST signal is active at low level. For a similar reason, the SWD and PTE30 terminals are disabled in autonomous mode. Finally, to support the underwater operation, the positive terminal of the power supply includes a general purpose rectifier diode.

Assuming that while in autonomous mode the battery always consumes energy, a new event is defined: the increase of the available energy in the battery. Based on that, it is proposed to evaluate the evolution of the battery energy and use an increment of the energy to trigger the switching from autonomous to supervisor mode.

### 3.1. Operation Modes

Figure 3 details the operation modes for the device. The core of this sensor node is a microcontroller that manages several on-board instruments, as well as everything related to energy management and on-error recovery tasks.

It is expected that most of the time the sensor node will be in autonomous operation mode, i.e., executing an experiment. Based on the scheme presented in Figure 1 and the operation modes in Figure 3, an experiment has three phases: START, WORKING, and END (see Figure 4).

The START and END phases require a short execution time in comparison with the one of the whole experiment. While in the START phase, the sensor node is configured, and the END phase basically waits for collecting data and results. Figure 5 illustrates the three phases in terms of execution time and the microcontroller working mode. In this figure, it can be observed that the WORKING phase is a sequence of run and sleep steps following the execution scheme described in Figure 1. In the START and END phases, the sensor node terminals are accessible, while in the WORKING phase its terminals are disabled.

The finishing of the WORKING phase may be due to three causes, becoming the device in one of the following three possible phases:*Correct end* (OK): The device ends the programmed experiment successfully and it has stored the results. Then, the device will be ready for providing the data to an external system or user.*Wrong end* (ERROR): This happens when the device aborts the experiments due to errors or failures. In this case, it is necessary to store debug information that might be later recovered.*Forced stop* (DEBUG): The device left the working phase because an external system or user has ordered to abort the experiment. In such a case, it is necessary that the device becomes under control of an external system or user, which could, eventually, recover data or debug information, or even reprogram and restart the device.

The first two phases belong to the normal operation of the sensor node itself. In these both phases the device terminals become immediately enabled. The last phase is the only one that involves an external action to trigger the internal execution of the routines for enabling the sensor node terminals.

### 3.2. Battery Powered System

The main feature of a traditional direct current (DC) power supply system is to provide a constant output voltage. However, the batteries provide a voltage depending on their stored energy. Figure 6 illustrates the discharge voltage of the UMAC040130A003TA01 from Murata Manufacturing Co., Ltd. under several constant currents.

In this research, for simulation purposes, the battery model presented in [33] for Li-Ion has been used. Table 2 enumerates several characteristic parameters of the sensor node battery. The discharge curve of a battery has three well-defined sections. In two of them, at the beginning and at the end, its slope is steep. The central section has a low slope and it is defined in most of the 60% of the discharging curve. Another intrinsic property of any battery is its power losses, which are minimized by design. This effect is modeled as a resistive load between terminals. Nevertheless, given a charged battery, the discharge from its losses is negligible compared to that from its applications.

Moreover, the working cycle of the battery should be understood, in this context, as a complete process for charging and discharging the device. In addition, it must be taken into account the characteristics deterioration with the number of work cycles. This makes unsuitable the use of fixed references to determine the state of these batteries.

From the point of view of the power consumption of the underwater sensor node, in Figure 5, it can be observed the microcontroller several run and sleep steps over time. The power consumption of the microprocessor in running mode is determined by the required hardware modules to execute the measurement specified in the experiment schedule. On the other hand, the sleeping mode of the microcontroller is used to save energy among measurements. The difference in power consumption between both microcontroller modes is greater than two or three decades. For example, the underwater sensor node used in this research runs in Very Low Power Run (VLPR) and sleeps in Very Low Leakage Stop (VLLS). On average, the current consumption in run mode is around 1 mA and the current in sleep mode is lower than 1 μA.

Based on previously considerations, Figure 7 shows the expected behavior in the battery output voltage when the underwater sensor node executes an experiment of two measurement iterations. The battery output voltage is decreased when the underwater sensor node goes into VLPR. Switching the microcontroller working mode implies that the output voltage of the battery increases its value. The output voltage stabilization for the Sleep state requires some time interval as shown between points labelled as C and D in Figure 7.

Obviously, the battery reduces its output voltage depending on the current demand and the elapsed time, e.g., at the beginning of the Exe state, labeled as A in Figure 7. As most of the experiment the microcontroller is sleeping, the battery output voltage (see Figure 7) is not sensed all the time. Therefore, the observable values from the microcontroller are those points in the curve between points labeled as A and C.

### 3.3. Trigger Latency

It has been assumed that a system user or designer might order to switch from autonomous to supervisor mode. In such a case, this is considered as an arbitrary external event. From the sensor node point of view, this means that the triggering arises at any time and during the execution of the experiment.

As the battery voltage is only sensed during the Exe state, the expected time to gain the control of the underwater sensor node is lower than the iteration time *T*. The condition to meet this upper limit is that the energy injected to the battery should be at least equal to the energy consumption throughout the last iteration of the experiment; otherwise, the triggering is not guaranteed.

## 4. Software Approach

Based on the behavior presented in Figure 7 and selected a sampling point between A and C, a first simple software solution can be proposed that consists in checking the power supply voltage at two identical points between consecutive iterations. This would result in executing the following operations:

**if** $VCC(t_{B_{i}})$ < $VCC(t_{B_{i+1}})$ **then** go to supervisor;

**else** continue in autonomous;

However, a problem arises when the temperature of the battery changes. Li-on batteries reduce its stored energy when the temperature decreases. Similarly, if the temperature increases, the available energy increases. For instance, this effect reduces the discharging time from 60 min at 25 °C to 52 min at 0 °C under a constant discharge current of 3 mA in a UMAC battery. This represents a 14% of variation in terms of stored energy, i.e., 0.4 mAh in terms of electric charge.

Therefore, this first and simple solution is only applied when the temperature does not change between measurements. Nevertheless, the ocean water has a temperature range between −2 °C and 35 °C on its surface [34]. The water temperature at deep ocean, under 200 m, is on average at 4 °C. The variation of this temperature also is affected by the solar radiation, which effects are greater when closer to the ocean surface. In general the variation due to the sun is near to 3 °C every 12 h. However, other critical events in the ocean produces greater changes, e.g., some researches [35] have even reported a variation of 14 °C in 3 min.

In conclusion, using this simple solution could lead to a glitch in the sensor node software under extreme weather scenarios.

### 4.1. Battery Status

As it is known, in order to determine correctly the energy stored in the sensor node battery, it is mandatory to measure its voltage, current or load, and the temperature. Measuring the current from the point of view of the sensor node is not viable. However, the battery energy level is always tracked by its charger when the device is programmed. Using this information to compute the discharging model is proposed.

From a practical point of view, the complexity of the battery model has to be taken in consideration. Basically, it is a set of discharging curves like those shown in Figure 6, where its input variables are the battery voltage, the time and the temperature. In addition, it has to be also taken into account the on-board computation limitations. An ultra-low power microcontroller has been required and, as an ARM M0+ was chosen for the application, this microcontroller does not incorporate a floating point unit.

On the other hand, the sensor node is a deterministic system. It helps to estimate the discharging behavior based on the power consumptions of the experiments programmed. The sensor node performs a set of actions (measurements and computations among others) following the experiment programmed. Each action can be characterized in terms of power consumption.

Therefore, the set of curves has been reduced for determining a suitable triggering event. In addition, instead of using complex models, those curves have been linearized using a piecewise function. Figure 8 presents the battery discharge piecewise model versus the real measurements at 25 °C for a constant current of 3 mA. The piecewise function used in this model is composed of 40 segments. The maximum error $V_{Epiece}$ between the piecewise model and the real measured values is 1 mV in absolute values. It is remarkable that the error in the battery nominal zone (time between 250 and 2700 s) is lower than 0.5 mV in absolute terms as shown in Figure 8b.

Another important design issue to take into consideration is the battery aging. It is well known that batteries degrade their characteristics with the number of charge–discharge cycles. For instance, the technical documentation of the battery used in this research indicates that after 5000 cycles its full charge capacity is at least an 80% or greater. However, the cycle counting alone is not enough to model the battery aging behavior. Other variables such as the working temperature and the charging–discharging currents reduce the so-called service life. Moreover, in general the sensor nodes never complete a charge–discharge cycle.

Because of these factors, the practical usage of a battery makes mandatory to measure its real status executing a complete charge–discharge cycle to evaluate the current battery degradation. This may be achieved charging the battery and then reproducing one of the discharging curves by measuring the voltage and the elapsed time. Therefore, the question is how often this verification should be run. For example, if we assume a linear degradation for the battery used in this paper, the maximum stored energy decreases 1% every 250 cycles.

If we assume that the sensor node executes the battery aging evaluation, some considerations must be taken into account. First of all, the sensor node should stop its measurement schedule during the checking time. This procedure is composed of several steps. In the first one the full charge is reached. In the next one, the battery is discharged during 1 h with a 3 mA current and the voltage and the temperature are measured. In the last one, the battery is recharged before continuing the device with its schedule.

An important issue to consider is that in the first and last steps it is required to control an external source power. Moreover, as the sensor node cannot ensure its own temperature, the acquired voltage curves are affected by temperature disturbances.

The complexity of fitting the acquired voltage and temperature data to the battery models in the sensor node are not practical from an implementation point of view. Therefore, in this research, the degradation of the battery is considered at the application level. At this high level, it is easy to compute the number of cycles to determine how long the device can be used assuming a maximum error. Reached this number of cycles, a possible action may be a maintenance operation in the laboratory or at sensor network level to check the battery. As a result, a new set of discharging curves might be uploaded to the sensor node.

### 4.2. On-Board Software Approach Details

It is pointed out in Section 3.2 that the battery voltage is measured between points A and C in Figure 7. However, it is also desirable to obtain measurements between points B and C. This is because measurements between A and B are more sensible to the temperature. In addition, a measurement close to C is highly dependent on the experiment iteration power consumption. Moreover, the underwater sensor node used in this research allows different measurements for each iteration. For example, this sensor node allows to measure the water turbidity in one iteration and an acceleration in the following iteration. Therefore, the best point to sample the battery voltage is exactly at B.

Figure 9 presents two consecutive iterations. At the first iteration, labeled as $T_{i}$, five turbidity measurements are obtained. The voltage drop owing to the LED turning on and off is clearly observed in Figure 9b. At the second iteration, labeled as $T_{i+1}$, a three axis acceleration measurement during 20 s is executed.

Each voltage drop in Figure 9c corresponds to the microcontroller storing acquired data into its flash memory. In this execution step, after the accelerometer initialization, the microcontroller hibernates between samplings. The accelerometer wakes up the microcontroller for storing the acquired data. That is the reason why the battery voltage rises above point B during the Exe step.

In both Exe steps, during the first second several tasks are performed. Initially, once the microcontroller wakes up into VLPR from the Sleep phase, the current demands produces a dramatic voltage drop. This consumption is also promoted by the sensor node system log, which stores some information in its flash memory. Then, some sanity checking is done. Finally, at point B, the temperature and battery voltage are measured. After this, in the Exe step, the programmed measurement is executed.

Once the temperature and battery voltage are obtained, the discharging piecewise linear function is selected from the memory. The difference between previous and current iteration voltages plus a predefined margin are compared with the piecewise linear prediction. In terms of microcontroller cycles, all the previously mentioned operations require less than 180 cycles, for the worst case scenario and without software optimization. The battery model is linearized by 10 piecewise functions, and each linear function is composed of 40 segments. In addition, the underwater sensor node used in our experiments measures its battery voltage and temperature for calibration purposes in each execution step. Therefore, it is not necessary to perform those measurements twice.

## 5. Experimental Results

### 5.1. Sensor Node under Test

As pointed out in Section 3, the presented method has been tested using the underwater sensor node previously presented [22]. Figure 10 shows its capsule, the main board and the battery. It uses a military grade aerospace heavy-duty aluminum capsule (see Figure 10a. The main board includes a microcontroller, three sensor measurement subsystems, an optional Crystal Oscillator at 32 KHz, an ISO 11874/11785 compliant wireless communication system and a battery.

Inside the capsule, all the electronic parts have been placed in a holder. It has been built using a 3D printer with Polylactic Acid (PLA) plastic. Figure 11 shows this holder and how it fits into the capsule. Note that once the main board and the battery are placed inside the holder, the remaining empty space between those parts and the holder is filled with resin. This is done in order to gain isolation as a second line of defense against humidity (see Section 2.1 for details).

In order to check the correctness of the proposal, the behavior of the underwater sensor node has been evaluated when the approach is applied. The precision measurement instrument B2912A from Keysight Technologies has been used to model and record every voltage and current of the sensor node and the battery.

The experimental platform that was built to carry out the experimental tests is shown in Figure 12. In addition, a connections box was developed to operate as a crossbar. It allows to automatize the swapping of the serial and parallel configurations of the instrument channels, the battery and the underwater sensor node.

### 5.2. Regular Operation

In this experiment, the aim is to check the behavior of the battery during a series of actions. The measured values are compared with the discharging battery model in Simulink Matlab. Table 3 presents the experiment program. It is composed of four Exe steps and three Sleep steps. The first Execution step E1 checks the sensor node integrity. Then, it hibernates for a period of 2700 s in step S1.

From Exe steps E2 to E4, a measurement of accelerations during 120 s is performed. Inside those steps, the sensor node hibernates 600 s in states S2 and S3. Finally, the device waits a data download in state W.

Figure 13 presents the real battery voltage behavior of the Table 3 experiment. The simulation of this experiment is also included. In Figure 13, the seven steps can be clearly seen. In this experiment, the sensor node is disconnected from its battery charger just at the start of the step E1. The noise produced by the crossbar switch from VCC to zero has not been included in the simulation, as it would produce a degree of uncertainty at the beginning of the simulation.

In this experiment, a light-emitting diode (LED) is also turned on and off (blinking) to produce a high current consumption and to observe the recovery effect. In a real application, this blinking does not take place, but it has been just established to analyze in detail the behavior of the equipment, as in the debugging of the device with this blinking it is possible to identify when the voltage measurement is actually done.

The simulation faithfully follows the measured voltage values. Since the battery models provide voltage values in average, it is not possible to detect the peaks of consumption. For example, as indicated in Section 4.2, the storage of data in the flash memory produces a great peak of consumption in comparison with the averaged value of power consumption given in the microcontroller documentation. Despite of this difference, the simulation does not differ so much in comparison with the voltages measured.

Finally, there is a small discrepancy between the simulated and the measured values for the consumption in sleep mode. The simulation underestimates slightly the discharging current with a low current demand, producing a divergence for long periods. This occurs because the simulation needs a feedback of the battery degradation due to the number of complete charging–discharging cycles performed. The discrepancy between simulations and measured values ($V_{ext}$) under 2 mV has been quantified for all the tests achieved in this research.

### 5.3. Starting point

Another interesting experiment has been to measure the behavior of the battery voltage with different starting voltages. Figure 14 presents the experiment proposed in Table 3 for two different starting points of the battery voltage. One has been fixed to 2670 mV as it belongs to the range of exponential behavior of the battery. The other one taken has been 2460 mV, which is its nominal value. In the following, the apostrophe symbol is used to denote the exponential behavior of the battery model, and when this symbol is not used the nominal behavior is considered.

An identical behavior can be observed in both measurements in Figure 14a, but with an evident difference in voltage level. In addition, the bigger the starting voltage of the battery, the clearer the recuperation effect in sleep steps.

Figure 14b details the initial behavior of E2${}^{\prime }$ on the left and E3${}^{\prime }$ on the right for the battery voltage when the experiment starts from 2670 mV. In this figure, it can be observed the voltage drop due to the three times blinking specified in E2${}^{\prime }$ in comparison with programming no blinking in E3${}^{\prime }$. In both cases, the selection of point B during the Exe step as reference to check the battery status ensures the non-influence of the programmed experiment. The battery voltage difference is in this case $V\left(B_{2}^{\prime }\right)-V\left(B_{3}^{\prime }\right)$ = 23 mV.

In a similar way, Figure 14c depicts the same behavior for the nominal voltage 2460 mV. The waveforms are identical, but the voltage difference is only $V\left(B_{2}\right)-V\left(B_{3}\right)$ = 3 mV.

### 5.4. Sensor Node Voltage Measurement

In this section, the internal voltage measurement performed by the microcontroller is evaluated. Nowadays, microcontrollers can measure their own power supply voltage and temperature. This may help to control the device status. Basically, they measure a Bandgap Voltage Reference (V${}_{ref}$) with their ADC. In the case of the MKL17Z256, the V${}_{ref}$ is trimmed at 1.195 V ± 1 mV. The 16-bit SAR-ADC effective bits are 12, and the following equation provides the power supply voltage:(1)$$V_{cc}=\frac{2^{12}\times V_{ref}}{V_{meas}},$$
where $V_{meas}$ is the measured value provided by the SAR-ADC when the voltage reference is selected. Note that, although $V_{ref}$ is a real number, $V_{meas}$ is an integer number in the [0,$2^{12}$] range. In this sense, the resolution of the SAR-ADC has an error of 0.65 mV and 0.4 mV when the battery voltage is 2.8 V and 1.8 V, respectively. Certainly, the effective bits may be enhanced by increasing the number of measurements and averaging obtained values, but the power consumption is also increased.

Finally, once the power supply voltage is determined, the ADC is calibrated and the temperature sensor is used. The system voltage and temperature log of the sensor node have been modified to sample the battery voltage on each action before and after its execution. For example, the E2 step samples the voltage at the beginning, after blinking the LED 3 times, also just after configuring the accelerometer and finally at the end of the acquisition stage. Figure 15 enumerates and locates the sampling points that were included in the sensor node program.

Figure 16 shows the voltages obtained using the microcontroller and the external values measured using the B2912A of the experiment. In a similar way to previous experiments, the values measured with the microcontroller and the external instrument follows the same tendency. However, there is an uncertainty which produces some slightly difference in the values. The root mean squared error (RMSE) and maximum absolute error (MAE) have been computed for all the samples. In Table 4, the obtained values are shown. The RMSE is 2.41 mV and the MAE is 1.99 mV when all samples are used. When only B points (samples 2, 6 and 9) are taken into consideration, those values are 1.51 mV and 1.23 mV. As it was expected, the selection of the point B as the location for measuring the voltage is the right place in terms of minimum error.

This experiment demonstrates that it is possible to analyze the behavior of the sensor node battery by sensing its voltage. It is also possible to take into consideration the regeneration effect through the internal voltage measurements in a real discharging scenario based on a practical point of view.

### 5.5. Triggering Event

The suitability of the triggering method which is proposed in this paper is evaluated in this experiment. In addition, the system is analyzed in the worst case scenario, which is assuming the occurrence of all the previously mentioned errors. This happens with
(2)$$|V_{error}|=V_{Epiece}+V_{ext}+V_{RMSE}=1+2.41+0.65=4.05\;mV$$

In the approach followed, the value of the error function in this worst corner case is added to the predicted battery voltage by the piecewise function models in programming time. When an internal measured battery voltage becomes over this dynamic threshold, the sensor node determines that a trigger event has occurred.

Figure 17 shows the behavior of the battery voltage for an experiment when charging currents of 0.5 mA and 1.0 mA are applied. In this experiment, after the third iteration plus some sleeping time (between 400 s and 450 s from the experiment start), a battery charger is connected to the sensor node (see labels $Tr^{1}$ and $Tr^{0.5}$ in Figure 17). The first consequence of this action is that the battery starts to charge energy, as the growing of its voltage on Figure 17 suggests. The trigger event arises at points $B_{4}^{0.5}$ and $B_{4}^{1}$. Then, the sensor node goes into the Wait state (with the LED continuously blinking), so the sensor node activates its terminals to get the supervisor operation mode.

An interesting effect of the charger connection is that it initially produces a visible increase of the battery voltage. Even though the charging current is only 0.5 mA, this initial voltage increase is about 8 mV, which represents the double of the accumulated error of the worst corner case.

## 6. Discussion on the Approaches

For the three possible methods, several advantages and weaknesses are presented in Table 1. In the following, the practical consequences of their usage are discussed in detail from a point of view of an ultra-low power underwater sensor node design.

### 6.1. Wireless Wake up Radio Receiver

First, this kind of wake up device requires complex circuits, protocols, and voltage adaptation in comparison with other solutions for a similar purpose. The higher the complexity of the circuits, the bigger the probability of a failure. In this discussion, this potential problem is ignored only in order to clarify the others mentioned.

Table 5 presents several recent research and industrial wake up receivers. Nowadays, the trade-off between power consumption and sensitivity is the design key, where the electronics researchers and industry focus their efforts. In the application under study, the parameter sensitivity will not play an important role, since it is assumed that the trigger wake up event will come up in controlled laboratory conditions. However, the power consumption of the wake up radio receiver in the idle or quiescent mode introduces a further loss of battery energy.

Other practical and very common problem in these approaches is that, in order to reduce their power consumption, the authors reduce the power supply voltage. However, those voltages do not fit within the operating voltage ranges of commercial batteries and microcontrollers. Therefore, it is mandatory to include in the sensor node a low-dropout (LDO) linear regulator, increasing moreover the power consumption of the whole system. For instance, if a TPS714xx from Texas Instruments is chosen, its quiescent current is 3.2 μA, which should be added to the second column of Table 5. In some cases, the LDO power consumption is greater than the wireless wake up unit.

In addition, the wake up radio circuit requires to use an extra interrupt pin of the underwater sensor node microcontroller. Therefore, the total power consumption of the wireless wake up solution will be always worse in terms of energy consumption and latency than the interrupt pin approach.

### 6.2. Pin Interruption

The most intuitive solution for the application considered is, in principle, to enable a microcontroller interrupt pin as input and activate its interruption software module. Due to the high water conductivity, the input pin circuitry of the microcontroller has to include a pull-down resistor and also there must be a mechanism for detecting the rising edge of the wake up input signal for triggering the interruption. Turning on an input pin for generating a microcontroller interruption does not only require the input amplifier of the pin pad and its pull-down—it also demands the activation of its the interruption module, usually called nested vector interrupt control (NVIC). In addition, it is also mandatory activate all the circuits to physically route the interrupt signal from the input pin to the NVIC. This infrastructure is called low leakage wake up unit (LLWU). In summary, the circuits associated to the pin interruption management introduce an additional energy consumption cost that must be evaluated. Its underestimation is a very common mistake when designing with ultra low power requirements.

In general, this information is not present or it is very confuse on the technical documentation of the devices. In the case of the MKL17Z256 microcontroller, under test in this research, its technical documentation provides some corner cases and also gives a very broad range of current consumptions. For example, this documentation indicates that the input leakage current is 25 nA and also that the microcontroller requires a maximum of 289 μA to run in very-low power run (VLPR) mode.

In order to evaluate the real energy consumption, several corner cases have be chosen based on the usage requirements of the underwater sensor node. Those corner cases are divided into two microcontroller modes. In this research, VLPR was chosen as pointed out in Section 3.1. The microcontroller sleeping is done in mode 1. The regular wake up procedure of the system is done by its low power timer (LPTMR) using an internal 1 kHz RC oscillator. Finally, the current consumption is evaluated when the internal analog to digital converter (ADC) is used for measuring the battery voltage and the internal temperature sensor. Table 6 presents the measured current consumptions for the corner cases established. Those values are measured using the Keysight Technologies B2912A precision source and measure unit (SMU).

As mentioned in Section 3.2, the current consumption difference between VLPR and Sleep1 modes is at least R0/S0 = 388 times bigger. The sensor node goes into Sleep mode 1 and also activates the LPTMR to wake up after the specified time. This also implies the usage of the NVIC unit. The LPTMR module was chosen because it has the lowest power consumption from all the available modes (LPTMR, TMP, PIT, or RTC). For the measurement task only, it was obtained that the pin interruption consumes 51 nA.

### 6.3. Software Approach

The microcontroller of the underwater sensor node uses an ARM M0+ as CPU. This processor is capable of executing an instruction per cycle in most of the cases. Given a software implementation which requires 180 cycles (see Section 4.2) running at 8 MHz in VLPR mode, the electric charge required by the software approach can be calculated as
(3)$${EC}_{SA}=\frac{Cycles}{{VLPR}_{freq}}\times \frac{I_{VLPR}}{1\;hour}=\frac{180}{8\times 10^{6}}\times \frac{244\times 10^{-6}}{3600}=1.525\;\mathrm{pAh}$$

### 6.4. Comparisons and Discussion

In order to compare the practical solutions, a set of typical underwater sensor nodes scenarios have been taken into consideration as follows:Case 1—Aquaculture feeding process: It consists in measuring the water turbidity of a tank or an offshore ocean cage every 24 h during not longer than 10 min [36].Case 2—Wave and tides recording: In the scenario, the hydrostaticpressure is measured during 15 s every 12 min [37].Case 3—Wave and tides recording: In similar a way to case 2, butthe measurement is executed 15 s every 30 s.Case 4—Fish physical activity: This device measures the acceleration of the fish operculum during 2 min every 15 min [38].

Table 7 summarizes the run and sleep times, and its ratio for a single day. The iteration period T of the experiments may go from 1 day to 1 second. The number of iteration periods executed in a day is equivalent to the number of executions of the SA approach per day. The working days is computed for a given 3000 μAh battery. The last five columns of this table present the equivalent μAh required extra electric charge, when the WM, PI and SA approaches are applied. From the wireless approaches studied (see Table 5), we have chosen the best in terms of minimum power consumption [30]. Although this WM solution requires a power supply voltage of 1.0 V, and it is mandatory to use a step-down voltage converter, we assume in our study that its power consumption is negligible in comparison with the WM module power requirements. This defines a best case scenario for the WM approach.

The iteration time T of the experiments goes from 1 day to 2 s. The working phase duration of the proposed experiments/cases are computed without including any method for gaining the control of the standalone sensor node. The cases studied for the given battery are to be working from 27.44 days up to just 10.56 h.

The WM always requires 7.13 times more electric charge than the PI approach. For instance, in Case 1 the equivalent extra energy required is 5713.49 μAh for the 27.44 days duration. In case of using the PI approach, the required energy is 800.52 μAh. Those requirements are very significant if the battery capacity is taken into consideration. This represents 190.45% and 26.68% of the total available energy. The WM approach usage requires the usage of a greater maximum capacity battery, in particular at least a 8.72 mAh battery. The PI approach also requires 3.8 mAh.

The minimum power consumption of the WM is located in Case 3. The reason is that the run to sleep ratio is 1. In this case the WM power consumption only needs 46.25 μAh and this represents the 1.54% of the whole battery capacity. The PI approach only requires a 0.22% in similar conditions.

From these results we may conclude that the WM approach can be applied to a ultra-low power consumption if the sleep to run ratio is lower than 6.5 to guarantee the minimum requirement of the 9.96% of the battery capacity. In case of using the PI solution, this ratio can be greater, in comparison with the WM, but not greater than 47 to keep the PI energy requirement under the 9.66% of the battery capacity.

On the other hand, the energy requirements of our proposal were evaluated following Equation (Equation 3). The equivalent consumption per hour is in the picoamperes (pA) scale. However, the PI approach requires 51 nA in the same conditions. The difference is greater than four orders of magnitude. Therefore, the comparison of those solutions depends only on the number of executions and how long the sensor node could operate, in terms of days. Obviously, the worst scenario for the SA solution proposed is the Case 3 with 43,200 repetitions of the battery sensing task. In this worst case the required energy is only 29.07 nAh. This consumption is negligible in comparison with the PI approach that requires 6.48 μAh in the same conditions. This poses 222.9 times more energy.

Assuming an extra current consumption of 51 nA when the sensor node activates an input interrupt pin in Sleep mode, the pin solution introduces more consumption when the Sleep step takes longer than 8.54 nAs/51 nA = 167.45 μs. This time determines the sleeping time that describes when using the methodology proposed becomes better than a PI solution in terms of power consumption.

In terms of latency, the longer the iteration time, the faster the WM and the PI approaches. A sensor node using our proposal requires no longer than 1 day to switch from standalone mode into debug mode. However, our proposal does not require more energy like the WM and PI solutions do. The results demonstrate that there exist a trade-off between iteration time and the energy required. In addition, we need to remember that the senor node is in stand-alone mode executing a programmed schedule. Therefore, it is easy to determine the remaining time to gain the control once the sensor node is plugged to its charger.

Another issue to consider is the fact of opening of the sensor node capsule to use its charging terminals (see Figure 2). Although the sensor node was in standalone due because its wireless communications were faulty and the sensor node is in the laboratory, it is possible to use a wireless charger. For example, the TMS37157 from Texas Instruments allows to be charged with a rate up to 80 mA. In order to keep the ultra low power condition of the SA, the TMS device must be configured in battery-less mode and its CPU interface must be disabled during the Working phase of the experiment.

## 7. Conclusions

Sensor nodes used in aquaculture applications need to be usually completely isolated from the environment. Sometimes, it is mandatory a communication channel for taking control of the isolated sensor node. In this paper, a method for taking the control based on sensing the sensor node battery has been proposed. As the device, in its normal operation mode, is completely isolated from outside, the procedure for gaining control of the sensor node to decide when this isolation should be canceled is based on measuring the battery voltage. When a supervisor needs to bring the sensor node out from isolation, the operator only needs to charge the battery using a standard charger or programmer. This causes a triggering event which is identified by a software procedure developed for this purpose. On the other hand, the battery regeneration effect and the periodical operation of the sensor node are taken in consideration and discussed in detail to model correctly the whole system.

Simulations have been carried out in order to study the operation of the method proposed, analyze its behavior using a set of practical corner cases, and establish its feasibility according to the application in which the sensor node is to be used. Those simulations have helped to build a piecewise linear function set depending on the voltage and the temperature. The piecewise linear model has been evaluated with real measurements using a high precision measurement instrument. The errors due to the modeling and the internal voltage measurements have been discussed and quantified in detail based on different practical examples using an underwater sensor node prototype, demonstrating the usefulness of the proposal.

Finally, the proposal has been compared in detail with similar state-of-the-art wireless solutions and the classic interrupt approach. From their comparisons, it has been determined that the proposed software approach is better, in terms of power consumption, than the pin interruption approach when the sensor node requires a hibernation time longer than 167.45 μs. Finally, our approach demands two orders of magnitude less energy than the best solution found in the literature.

## Figures and Tables

**Figure 1 sensors-21-04660-f001:**
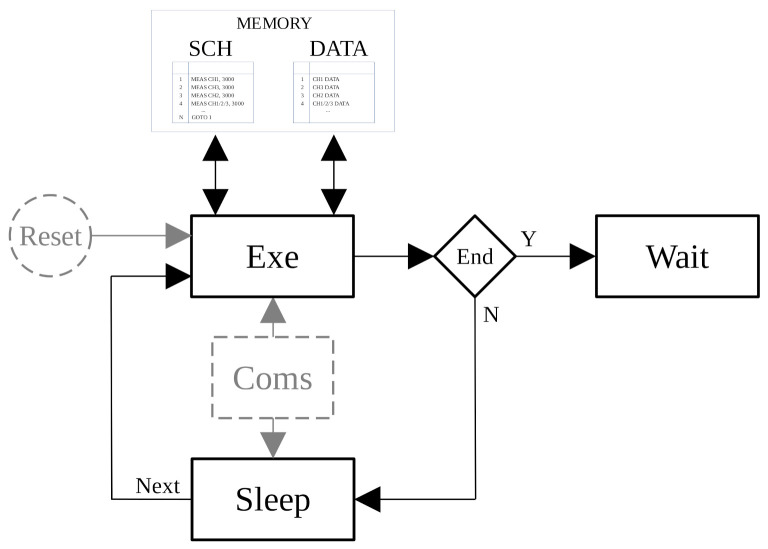
Basic execution scheme of a sensor node.

**Figure 2 sensors-21-04660-f002:**
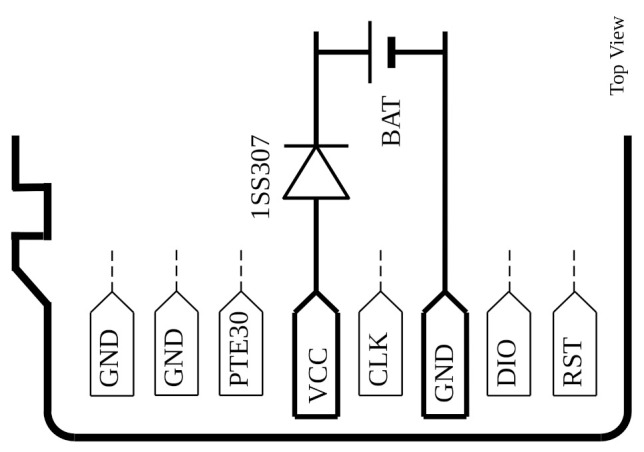
Connector used in the underwater sensor node with identification of terminals [22].

**Figure 3 sensors-21-04660-f003:**
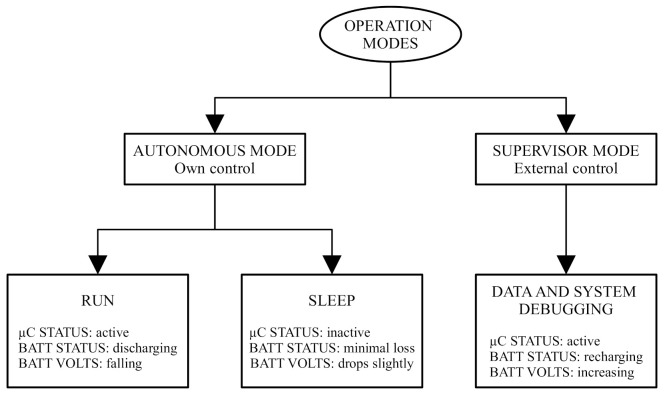
Diagram of the modes, states, and phases of operation existing in the sensor node.

**Figure 4 sensors-21-04660-f004:**
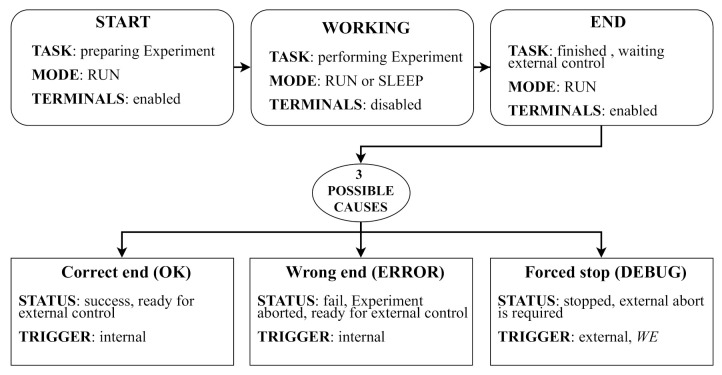
Detail of the experiment phases, sensor node status, and operation modes triggering mechanisms.

**Figure 5 sensors-21-04660-f005:**
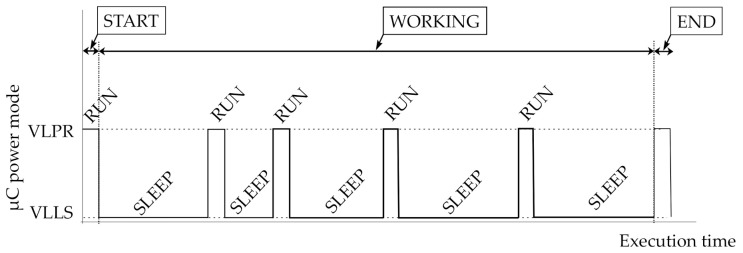
Sensor node phases for a generic experiment. Relationship with the microcontroller power modes: very-low-power run (VLPR)—corresponding to the RUN state—and very-low-leakage stop (VLLS)—the SLEEP state.

**Figure 6 sensors-21-04660-f006:**
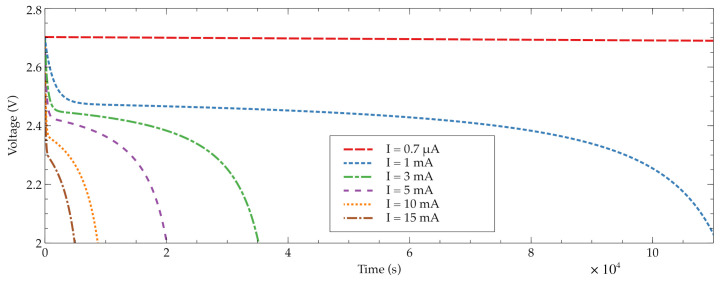
Voltage behavior of a battery when discharged at several constant currents.

**Figure 7 sensors-21-04660-f007:**
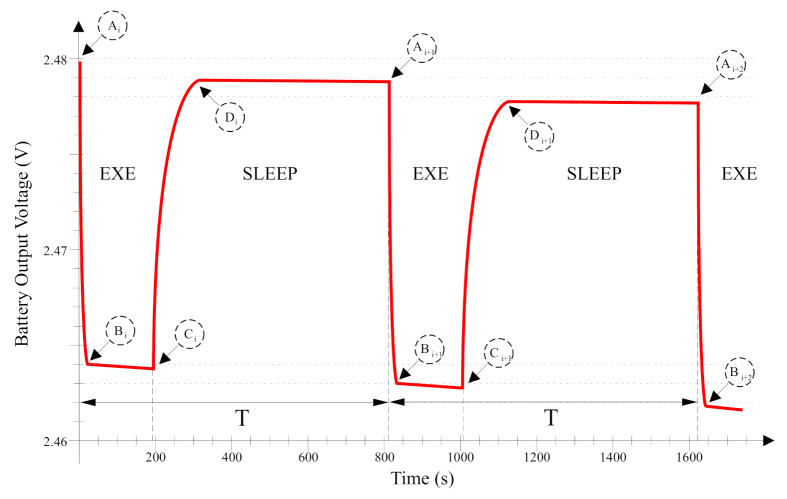
Battery voltage behavior for two iterations and relevant points: (**A**) Starting the Execution step. (**B**) Voltage stabilization for the Execution step. (**C**) Recovery phase. (**D**) Dropping phase sleeping till the next wake up.

**Figure 8 sensors-21-04660-f008:**
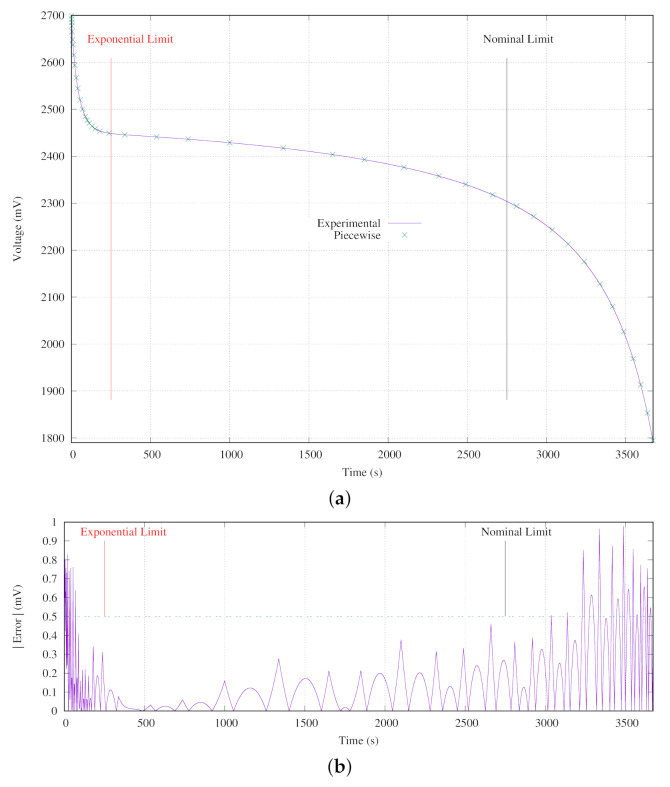
Battery model: (**a**) Experimental measurements and 40 sections piecewise function. (**b**) Absolute error between measured data and piecewise function.

**Figure 9 sensors-21-04660-f009:**
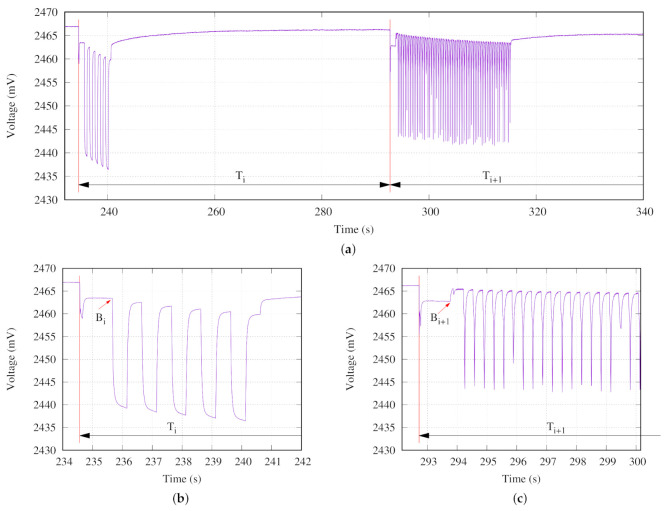
Two consecutive iterations experiment: (**a**) Measured battery voltage, (**b**) turbidity measurement, and (**c**) acceleration measurement.

**Figure 10 sensors-21-04660-f010:**
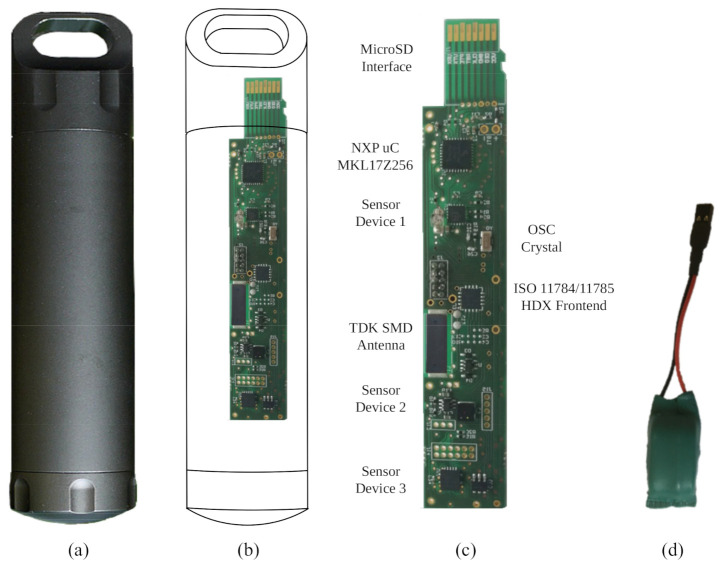
Underwater sensor node: (**a**) capsule, (**b**) comparison of capsule and main board sizes, (**c**) main board details, and (**d**) battery.

**Figure 11 sensors-21-04660-f011:**
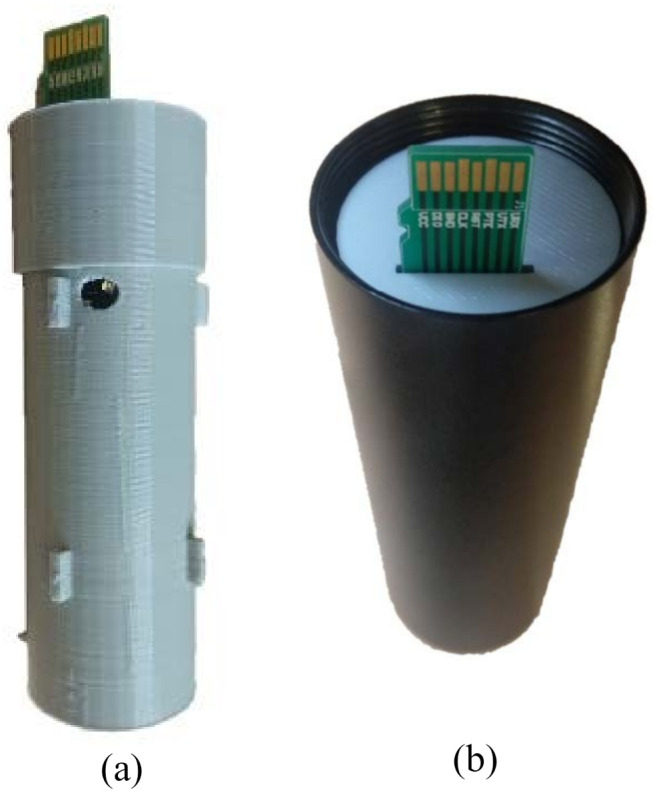
Underwater sensor node construction details: (**a**) PLA holder for node capsule with the main board and battery, and (**b**) holder inside of the container.

**Figure 12 sensors-21-04660-f012:**
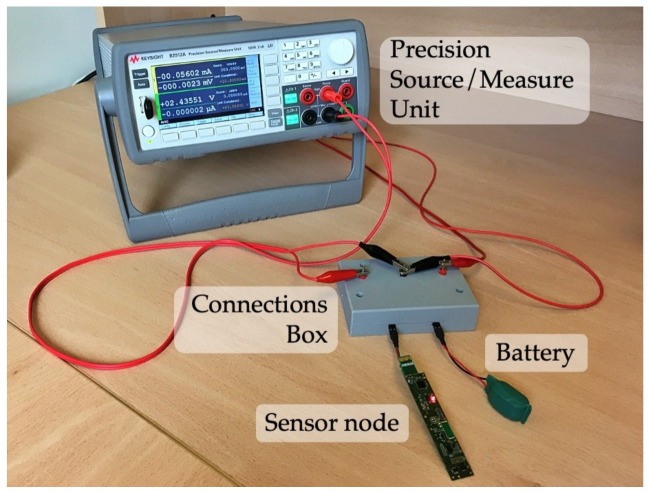
Experimental platform built for tests.

**Figure 13 sensors-21-04660-f013:**
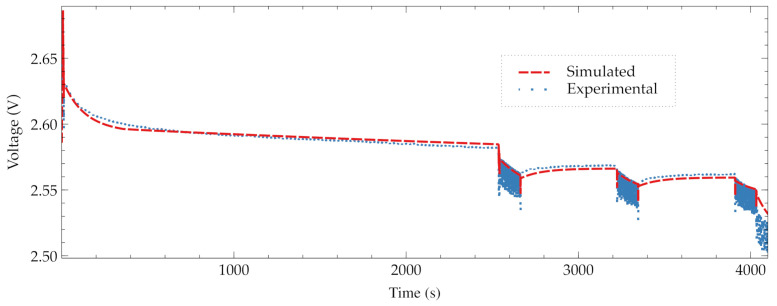
Comparison between simulation and real battery voltage for an experiment executing three identical measurements of accelerations.

**Figure 14 sensors-21-04660-f014:**
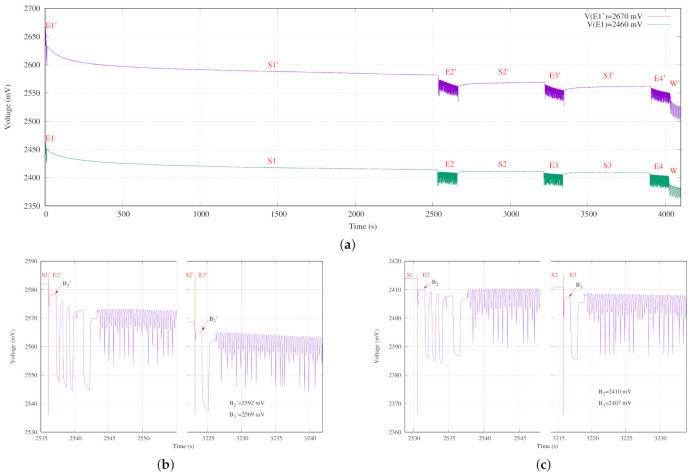
Two experimental tests — behavior comparison for different initial charge levels. (**a**) Comparison between tests. (**b**) Detail of recovery effect in the sensor node starting with a higher level of charge. (**c**) Detail of recovery effect in the sensor node with a lower level of charge.

**Figure 15 sensors-21-04660-f015:**
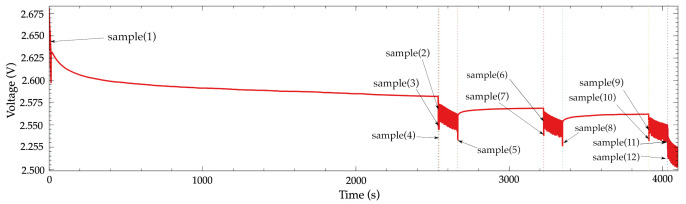
Samples for behavior study. Sampling instants of the internal voltage measurements.

**Figure 16 sensors-21-04660-f016:**
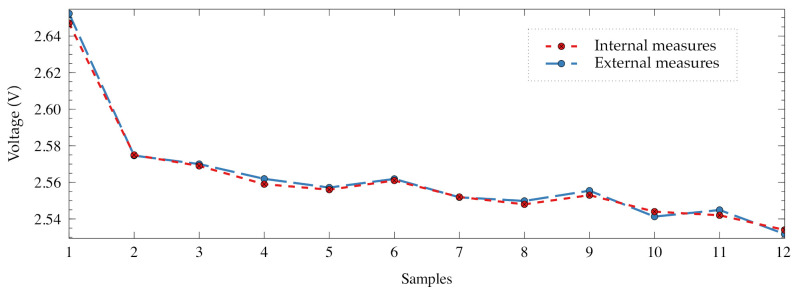
Samples for studying the behavior. Comparison between internal and external voltage measurements.

**Figure 17 sensors-21-04660-f017:**
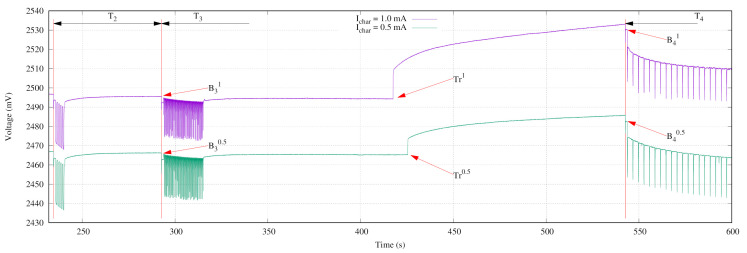
Experiment to switch the autonomous mode to supervisor mode using different charging currents.

**Table 1 sensors-21-04660-t001:** Advantages and weaknesses of the trigger mechanisms switching to supervisor mode.

Method	Extra Area	Physical Problems	Wake up Response	Power Consumption	Literature
Wireless (WM)	Yes	Antenna Orientation	Middle	High	[28,29,30]
Pin Interruption (PI)	No	Manipulation	Fast	Ultra Low	[31,32]
Software (SA)	No	Isolated	Slow	Cycles Dependent	Our proposal

**Table 2 sensors-21-04660-t002:** Characteristic parameters for the underwater sensor node battery.

Parameter	Value
Nominal Voltage	2.3 V
Energy Charge	3.0 mAh
Cut-off Voltage	1.8 V
Full Charged Voltage	2.7 V
Internal Resistance	0.8 Ω

**Table 3 sensors-21-04660-t003:** Experiment program to evaluate the battery model.

#	Led Blink	Action Description	Step
00	x1–x1	Experiment start	E1
01	x1–x1	Sleep 2700 s	S1
02	x3+x1–x1	Meas Accel 120 s	E2
03	x1–x1	Sleep 600 s	S2
04	x1–x1	Meas Accel 120 s	E3
05	x1–x1	Sleep 600 s	S3
06	x1–x1	Meas Accel 120 s	E4
07	x1@all	Blink LED and wait for data download	W

x1–x1: 1 blink before and after execute action. x3+x1–x1: 3 fast plus 1 blink before and only 1 blink after execute action. x1@all: blink forever.

**Table 4 sensors-21-04660-t004:** RMSE and MAE of the battery voltage with external vs. internal measurements.

Parameter	Value (mV)	B Point Value (mV)
$V_{RMSE}$	2.41	1.51
$V_{MAE}$	1.99	1.23

**Table 5 sensors-21-04660-t005:** Comparison of recent wake up receiver (WUR) circuits.

Voltage (V)	Sleeping (μA)	Consumption (μW)	Frequency	Area (μm × μm)	Sensitivity (dBm)	Year and Reference
0.5	600.0	300.0	928 MHz	2000 × 1000 (Die)	−103 dBm	2021 [29]
1.2	0.440	0.528	868 MHz	634 × 391 (Die)	−62 dBm	2020 [28]
1.016	0.364	0.370	<200 MHz	370 × 250 (Die)	−79 dBm	2019 [30]
2.0	0.300	0.600	134.2 kHz	4000 × 4000 (VQNF)	–	TMS37157
2.4	1.37	3.288	134.2 kHz	2700 × 2700 (QFN)	–	AS3930
1.8	600.0	1080	960 MHz	3000 × 3000 (MSOP)	–	Si4012
1.6	600.0	960	<960 MHz	3850 × 3850 (QFN)	−118 dBm	SPIRIT1

**Table 6 sensors-21-04660-t006:** Current consumptions measured on the MKL17Z256 in several working modes at 8 MHz or sleeping and required internal modules.

Corner Case	Consumption	CPU Mode	Additional Modules
S0	629 nA	Sleep1	LPTMR + LLWU
S1	681 nA	Sleep1	LPTMR + LLWU + Pin Int.
R0	244 μA	VLPR	Alone CPU
R1	566 μA	VLPR	ADC Vcc and Temp

**Table 7 sensors-21-04660-t007:** Required electric charge for given underwater measurement scenarios.

Case	Run (s)	Sleep (s)	Sleep/Run (Ratio)	T (s)	1 Day/T	Working (Days)	WM ${}^{1}$		PI ${}^{2}$		SA ${}^{3}$ (nAh)
							(μAh)	(%)	(μAh)	(%)	
1	600	85,800	143.0	86,400	1	27.44	5713.49	190.45	800.52	26.68	41.84 × 10${}^{-3}$
2	1800	84,600	47.0	720	129	10.07	2068.25	68.94	289.78	9.66	1.98
3	43,200	43,200	1.0	2	43,200	0.44	46.25	1.54	6.48	0.22	29.07
4	11,520	74,880	6.5	900	96	1.64	298.82	9.96	41.87	1.40	0.24

${}^{1}$: $I_{WM}$ = 367 nA using [30]. ${}^{2}$: $I_{PI}$ = 51 nA. ${}^{3}$: $EC_{SA}$ = 1.525 pAh.

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
