# Peer review of "A New Method for Gaining the Control of Standalone Underwater Sensor Nodes Based on Power Supply Sensing"

_sensors, 2021, doi:10.3390/s21144660_

Round 1
Reviewer 1 Report
The content of this manuscript is much improved, and the comparison among wireless wake-up radio receiver, pin interruption, and software approach, is a good plus.
- The term “wake-up” in the title is a little misleading because this paper only addresses state change by periodical pulling (duty-cycling), and it does not involve the activation of a sensor node from sleep. Please consider refining the title if possible.
- Although discharging function per temperature is measured, it is possible that this property changes with battery aging. Please discuss this.
- English writing needs improvement.
Author Response
Please, see the attatched PDF file.

Reviewer 2 Report
The related work commented in the introduction needs more deep, e. g. paragraph 73-80 references 6 articles and it does not cover how each different paper deals with the problem of taking control of the isolated node. In addition, no contrast or comparison is detailed in that section between existing approaches and the proposed one. Moreover, a related work section would be welcomed to write down all those matters.
Section 2.3 needs some references to check approaches that used those mechanisms. In wireless mechanism it is necessary to comment about what wireless technologies would be suitable.
Section 6. Is the power consumption for case 6.2 continuous or just instantaneous when the interrupt is launched? 6.2 and 6.3 cases both requires manipulation, and when you have the device in your hands what’s the point of measuring the energy cost of just an interrupt activation?
Although the presented approach shows a working solution it is not clear which improvements achieves compared with just a pin interruption because both requires manipulation of the sensor outside the water. Moreover, the presented approach it is not well compared with similar solutions in the related work
Minor comments
- Typo in abstract: line 11 “preformed”.
- Figures 1 and 2 should be larger, to allow text within them to be similar size than standard font size.
Author Response
Please, see the attached PDF file.

Round 2
Reviewer 2 Report
The author have adressed all the required changes